# Association of intraindividual differences in estimated glomerular filtration rates based on cystatin C and creatinine with dementia: A cohort study of the UK Biobank

Zhiyi Mao[1,2,◉], Yuwei Peng[1,◉], Ruilang Lin[1], Xinyue Guo[2], Xiaorui Cui[1], Yongfu Yu[1], Xueying Zheng[1,2]*

**1** Key Laboratory of Public Health Safety of Ministry of Education, NHC Key Laboratory for Health Technology Assessment, School of Public Health, Fudan University, Shanghai, China, **2** Department of Biostatistics, School of Public Health, Fudan University, Shanghai, China

◉ These authors contributed equally to this work.
* xyzheng@fudan.edu.cn

## Abstract

### Background

Dementia is a leading cause of cognitive decline, with Alzheimer's disease (AD) and vascular dementia (VaD) being the most common subtypes. The intraindividual difference between the estimated glomerular filtration rate based on cystatin C and creatinine ($eGFR_{diff}$) may serve as an indicator of the overall health status of an individual. However, the relationships between the $eGFR_{diff}$ and dementia risk, dementia subtypes, dementia-related neuroimaging changes, and cognitive functions remain unclear.

### Methods

This study analysed data from over 450,000 participants in the UK Biobank who were followed for up to 15 years. The estimated glomerular filtration rate based on cystatin C ($eGFR_{cys}$) and creatinine ($eGFR_{cr}$) was calculated using the CKD-EPI equation, and $eGFR_{diff}$ was defined as the difference between these values ($eGFR_{diff} = eGFR_{cys} - eGFR_{cr}$). Multivariate Cox regression models were used to evaluate the associations between the $eGFR_{diff}$ and all-cause dementia (ACD), AD, and VaD, whereas cross-sectional analysis were used to examine the relationship among the $eGFR_{diff}$, dementia-related neuroimaging changes, and cognitive functions.

### Results

Over a median follow-up of 13.5 years, 8,710 participants developed dementia, including 3,910 with AD and 1,893 with VaD. Each one standard deviation increase in $eGFR_{diff}$ was associated with a reduced risk of dementia, with hazard ratios (95%

**Data availability statement:** Data cannot be shared publicly because individual-level data deposition is explicitly prohibited by the UK Biobank Material Transfer Agreement and Research Ethics Committee approval (16/NW/0274). Data are available from the UK Biobank Access Team (contact via access@ukbiobank.ac.uk or https://www.ukbiobank.ac.uk/enable-your-research/apply-for-access) for researchers who meet the criteria for access to confidential data. The data underlying the results presented in the study are available from UK Biobank (Application ID: 98410).

**Funding:** This work was supported by the National Natural Science Foundation of China (grant No. 82273730 to YY), the Shanghai Municipal Natural Science Foundation (grant No. 22ZR1414900 to YY), and the Three-Year Public Health Action Plan of Shanghai (grants GWVI-11.2-XD10 and GWVI-11.1-01 to YY). The funders had no role in study design, data collection and analysis, decision to publish, or preparation of the manuscript. There was no additional external funding received for this study.

**Competing interests:** The authors have declared that no competing interests exist.

**Abbreviations:** ACD: all-cause dementia, AD: Alzheimer's disease, VaD: vascular dementia, eGFR: estimated glomerular filtration rate, eGFRcr: eGFR based on creatinine, eGFRcys: eGFR based on cystatin C, WMHs: white matter hyperintensities, UKB: UK biobank, MRI: magnetic resonance imaging

confidence intervals) of 0.92 (0.90–0.94) for ACD, 0.94 (0.91–0.98) for AD, and 0.90 (0.85–0.94) for VaD. A negative $eGFR_{diff}$ was associated with adverse neuroimaging changes, including lower total brain and gray matter volumes and higher white matter hyperintensities. Additionally, a negative $eGFR_{diff}$ was associated with poorer performance across multiple cognitive domains.

## Conclusion

A negative $eGFR_{diff}$ was associated with an increased risk of dementia, adverse neuroimaging outcomes, and cognitive decline. These findings suggest that the $eGFR_{diff}$ might be considered a potential associative indicator for dementia and cognitive impairment, suggesting potential clinical value in risk assessment and early intervention strategies.

## Background

Dementia is a complex neurological disorder characterized by progressive and irreversible cognitive decline. In 2019, the incidence of dementia was reported to be 82.9 cases per 100,000 individuals in the UK (ranging from 70.6 to 95.1) and 95.0 cases per 100,000 individuals globally (ranging from 81.6 to 107.9) [1]. The prevalence of dementia is projected to increase from 50 million cases in 2020 to approximately 152 million cases by 2050 [2]. Alzheimer's disease (AD) and vascular dementia (VaD) are the primary subtypes of dementia, accounting for 60% and 20% of all cases, respectively [3]. Given the significant personal and socioeconomic burden of dementia, early detection and intervention are crucial for slowing disease progression and improving patients' quality of life [4,5]. Identifying reliable clinical markers in high-risk individuals is essential for facilitating timely diagnosis and management.

Although several studies have explored the relationships between the estimated glomerular filtration rate (eGFR) and dementia or cognitive impairment, the findings have been inconsistent [6–10]. These inconsistencies may be attributed to the differential influence of non-renal factors on eGFR estimation methods, particularly when using serum creatinine ($eGFR_{cr}$) versus cystatin C levels ($eGFR_{cys}$) [11,12]. Given the variability between these two measures, the difference between the $eGFR_{cys}$ and $eGFR_{cr}$ ($eGFR_{diff}$) has been proposed as a novel clinical marker for dementia or cognitive impairment. Recent studies suggest that the $eGFR_{diff}$ is associated with various adverse clinical outcomes, including depression, heart failure, all-cause mortality, frailty, diabetic microvascular complications, cognitive decline, and motoric cognitive risk syndrome [13–21]. However, its relationship with specific dementia subtypes and subtle, preclinical dementia-related changes remains poorly understood. Understanding the potential role of the $eGFR_{diff}$ as an indicator of early neurodegenerative processes could provide valuable insights into dementia risk and facilitate its early detection.

To address these knowledge gaps, this study investigated the associations between the $eGFR_{diff}$ and the risk of developing all-cause dementia (ACD), AD, and

VaD in a large population-based cohort followed for up to 15 years. Given that the preclinical stage of dementia is characterized by subtle changes in brain structure and cerebrovascular pathology [22], this study hypothesized that $eGFR_{diff}$ serves as an early indicator of these dementia-related changes. To further explore this hypothesis, this study analysed associations between $eGFR_{diff}$, neuroimaging changes, and cognitive functions.

## Materials and methods

### Study design and population

This study utilized data from the UK Biobank (UKB), recruiting over 0.5 million participants aged 40–69 years from 22 centers across England, Scotland, and Wales between 2006 and 2010. At baseline, health-related data were collected through touchscreen questionnaires, verbal interviews, and physical measurements. Follow-up data were obtained through cohort-wide linkage to electronic health records. A total of 502,250 participants were initially enrolled. The exclusion criteria included withdrawal of consent, missing blood biochemistry data, self-reported or hospital-diagnosed cognitive impairment, dementia, or traumatic brain injury at baseline, and missing covariate data. The final study cohort included 458,668 participants (S1 Fig in S1 File). Neuroimaging and cognitive function analyses were performed in a subset of the baseline cohort who participated in the UKB imaging initiative [23]. During the imaging visits, brain magnetic resonance imaging (MRI) was performed using a 3 Tesla Siemens Skyra scanner equipped with software VD13 software and a standard 32-channel head coil. Imaging-derived phenotypes (IDPs) are generated via standardized image processing and quality control procedures [24]. Cognitive function was assessed via a touchscreen questionnaire [25]. After excluding participants with a history of cognitive impairment or dementia, 38,581 participants were included in the neuroimaging analyses, whereas 148,759 participants were included in the cognitive function analyses.

Ethical approval for the UKB study was granted by the North West Multi-Centre Research Ethics Committee (11/NW/0382). Informed consent was electronically obtained from all participants through a touchscreen. The current project was conducted under UK Biobank application ID: 98410. No additional ethical approval or patient re-contact was required.

### Data collection

**Assessment of eGFR differences.** Serum creatinine and cystatin C levels were measured via enzymatic and latex-enhanced immunoturbidimetric assays, respectively. The $eGFR_{cys}$ and $eGFR_{cr}$ were calculated using the 2021 race-free Chronic Kidney Disease Epidemiology Collaboration (CKD-EPI) equation [11, 26]. The $eGFR_{diff}$ was defined as the $eGFR_{cys}$ minus the $eGFR_{cr}$, where negative values indicated a lower $eGFR_{cys}$ than the $eGFR_{cr}$, and positive values indicated a higher $eGFR_{cys}$ than the $eGFR_{cr}$. Additionally, the $eGFR_{ratio}$, defined as $eGFR_{cys}/eGFR_{cr}$, was calculated for sensitivity analysis. On a basis of predefined thresholds, $eGFR_{diff}$ was categorized into three groups: 1) negative $eGFR_{diff}$ ($<-15$ ml/min/1.73 $m^2$); 2) midrange $eGFR_{diff}$ ($-15–15$ ml/min/1.73 $m^2$); and 3) positive $eGFR_{diff}$ ($\geq 15$ ml/min/1.73 $m^2$) [19,20,27]. The $eGFR_{ratio}$ was stratified into quartiles for sensitivity analysis.

**Outcomes.** The primary outcomes included incident ACD, AD, and VaD. Incident dementia was defined as the first hospital inpatient diagnosis of dementia or documentation of dementia as a contributing or underlying cause of death [28]. Both the inpatient datasets and death registers use the International Classification of Diseases (ICD) coding system. The UKB Outcome Adjudication Group compiled and validated ICD codes for dementia and its subtypes [29]. The dates of ACD, AD, and VaD diagnosis were considered the primary outcomes.

For neuroimaging analyses, IDPs were extracted from T1-weighted and T2-weighted fluid-attenuated inversion recovery (T2-FLAIR) MRI images, which was [23]. MRI was acquired a median of 4.7 years (IQR 3.9–5.5) after baseline, ensuring that the exposure ($eGFR_{diff}$) temporally preceded the imaging outcomes. Imaging markers indicative of brain health included total brain volume, total white matter volume, total gray matter volume, bilateral hippocampus volume, and white matter hyperintensity (WMH) volume. WMH volume was log-transformed because of its skewed distribution.

Cognitive function including executive function, verbal and numerical reasoning, working memory, complex processing speed, verbal declarative memory, and non-verbal reasoning was assessed via validated touchscreen-based cognitive tests [25].

**Covariates.** Demographic, socioeconomic, clinical, and genetic factors including ethnic background, sex, were considered. 13 modifiable dementia risk factors identified by Livingston et al., which account for more than 40% of dementia onset were also considered to avoid potential confounders [4]. Sociodemographic, lifestyle, medical, and medication data were collected at baseline via touchscreen questionnaires. Anthropometric measurements such as height, weight, blood pressure, and cholesterol levels were obtained. Participants with cholesterol levels exceeding 5.7 mmol/L and low-density lipoprotein (LDL) levels above 2.6 mmol/L were classified as having high cholesterol or high LDL. Education was categorized as "college or above," "high school or equivalent," or "less than high school" [30]. Smoking status was classified as "never" or "ever," and alcohol consumption was classified as "never," "current," or "current-excess" (more than 21 UK units) [4,31]. Physical activity was categorized by MET values, and socioeconomic status was estimated via the Townsend deprivation index [32,33]. Hearing and vision impairment were assessed by self-reported difficulties [34]. Diabetes, hypertension, depression, and obesity were identified through self-reports or relevant medications. To assess the effect of kidney function, $eGFR_{cr\_cys}$, was included as a covariate [11,26]. Genetic risk factors were evaluated via APOE genotypes and a non-APOE polygenic risk score (PRS) specific to Alzheimer's disease. The participants were categorized by APOE genotype and non-APOE PRS status. More details on the genetic methodology are available on the UK Biobank website(https://biobank.ctsu.ox.ac.uk/showcase).

## Statistical analysis

Continuous variables are expressed as the means with standard deviations (SD) or medians with interquartile ranges (IQR) depending on their distribution. Comparisons between groups were performed using Student's *t*-test or the Mann–Whitney U test, as appropriate, on the basis of data distribution. Categorical variables are expressed as counts (percentages) and analysed via the chi-square test or Fisher's exact test, as appropriate. To examine the association between $eGFR_{diff}$ and incident dementia, Cox proportional hazard regression was performed, adjusting for potential confounders, including ethnicity, sex, Cho, LDL, education, smoking status, alcohol consumption, physical activity, TDI, social isolation, hearing impairment, vision impairment, diabetes, hypertension, depression, and obesity. The participants were followed-up from baseline assessments until dementia diagnosis, death, or censoring of hospital inpatient records (October 31, 2022, for England; August 31, 2022, for Scotland; and May 31, 2022, for Wales). The proportional hazard assumption was assessed using the Kaplan–Meier curves and Schoenfeld residual analysis. The results are reported as hazard ratios (HRs) with 95% confidence intervals (CIs). Subgroup analyses were conducted to assess potential effect modification by APOE-genotype, non-APOE PRS, and physical impairment including hearing loss and vision loss. To investigate the associations between z scores of $eGFR_{diff}$ and neuroimaging outcomes as well as cognitive functions, multivariable linear regression analyses were performed, with adjustments made for the same covariates as in the Cox regression. The sensitivity analyses including the following: 1) multiple imputation with the default setting for missing covariate data [35], 2) exclusion of participants with an $eGFR_{cr}$ or $eGFR_{cys}$ less than 60 mL/min/1.73 m$^2$ due to the potential unreliability of their eGFR measurements [36], 3) testing other eGFR difference indices in place of the $eGFR_{diff}$ to ensure the reliability of the findings, 4) repeating the primary analyses within subgroups of participants with major pre-existing diseases, including diabetes, hypertension, and depression, and 5) lag-time analysis excluding participants diagnosed with dementia within 5 years post-exposure, assuming they may have had prevalent disease at baseline [37]. The reporting of our study adheres to the STROBE guidelines [38]. In addition, exploratory risk stratification and discrimination analyses were performed(details in S1 method in S1 File). All the statistical analyses were performed via R 4.0.3 software. A two-tailed *p*-value < 0.05 was considered statistically significant.

## Results

### Baseline characteristics

The final study cohort comprised 458,668 participants with a mean age of 56.5 years (SD = 8.09), of whom 54.3% were female. Table 1 summarizes the demographic and clinical characteristics of the study population. The participants with lower $eGFR_{diff}$ values were generally older and predominantly male. Health and social determinants also varied significantly across the $eGFR_{diff}$ groups. The negative $eGFR_{diff}$ group presented the highest levels of educational attainment, smoking and alcohol consumption rates, and TDI scores, with these levels progressively decreasing as $eGFR_{diff}$ increased. Additionally, this group demonstrated lower high-intensity but higher moderate-intensity physical activity, along with a higher prevalence of depression, obesity, and hypertension. The positive $eGFR_{diff}$ group presented the highest incidence of diabetes. LDL cholesterol levels were elevated in the midrange $eGFR_{diff}$ group, whereas vision impairment was most severe in the negative $eGFR_{diff}$ group ($p < 0.001$).

At baseline, the mean $eGFR_{cr}$ and $eGFR_{cys}$ values significantly differed across the $eGFR_{diff}$ groups. The mean $eGFR_{cr}$ values decreased with increasing $eGFR_{diff}$, whereas the mean $eGFR_{cys}$ values increased (both $p < 0.001$). A significant positive correlation was found between the $eGFR_{cys}$ and $eGFR_{cr}$ (R = 0.604, $p < 0.001$). The distributions of these measures are shown in S2 Fig in S1 File.

### Association of the $eGFR_{diff}$ with dementia

Over a median follow-up period of 13.5 years (interquartile range: 13.1–14.5 years), a total of 8,710 participants developed ACD, including 3,910 patients with AD, 1,893 patients with VaD, and 274 patients with frontotemporal dementia. Further analyses were not performed for frontotemporal dementia due to the low incidence rates. Kaplan–Meier survival curves and Schoenfeld residual analysis confirmed the proportional hazard assumption and revealed significant differences in outcome risks across the $eGFR_{diff}$ groups (S3 Fig in S1 File). Participants in the positive $eGFR_{diff}$ groups presented lower risks of developing ACD, AD, and VaD than the negative $eGFR_{diff}$ group (Table 2).

Specifically, for the incidence of ACD, the multivariable-adjusted HR was 0.88 (95% CI: 0.84–0.92) for the midrange $eGFR_{diff}$ group and 0.74 (95% CI: 0.65–0.86) for the positive $eGFR_{diff}$ group, compared with the negative $eGFR_{diff}$ group. A similar protective trend was observed for the incidence of AD, with adjusted HRs of 0.92 (95% CI: 0.85–0.98) in the midrange $eGFR_{diff}$ group and 0.80 (95% CI: 0.65–0.98) in the positive $eGFR_{diff}$ group. For the incidence of VaD, the adjusted HRs were 0.85 (95% CI: 0.77–0.94) for the midrange $eGFR_{diff}$ group and 0.67 (95% CI: 0.49–0.93) for the positive $eGFR_{diff}$ group. The dose–response relationships between the $eGFR_{diff}$ and the incidence of both ACD and AD primarily exhibited an L-shaped pattern, although nonlinearity tests revealed significant only in the incidence of ACD (Fig 1). For each 1 SD increase in $eGFR_{diff}$, the corresponding HRs (95% CIs) were 0.92 (range: 0.90–0.94) for the incidence of ACD, 0.94 (range: 0.91–0.98) for the incidence of AD, and 0.90 (range: 0.85–0.94) for the incidence of VaD (Table 2).

### Subgroup analysis and sensitivity analysis

Stratified analyses based on APOE genotype, PRS, sex and ethnicity (Fig 2) revealed that the associations between the eGFRdiff and incident dementia remained consistent across subgroups. However, the interactions observed between APOE genotype and ethnicity significantly modified the relationships between the eGFRdiff and the incidence of ACD, while ethnicity alone significantly influenced the association between the eGFRdiff and the incidence of VaD.

Sensitivity analyses demonstrated that the magnitude of associations remained stable after multiple imputations were applied for missing data (S1 Table in S1 File), excluding of participants with $eGFR_{cr}$ or $eGFR_{cys}$ values less than 60 mL/min/1.73 m$^2$ (S2 Table in S1 File), and when alternative eGFR difference indices were used in place of $eGFR_{diff}$ (S3 Table, S4 Fig in S1 File). Notably, the associations between the $eGFR_{cr}$ or $eGFR_{cys}$ and the incidence of dementia varied before and after adjustment for kidney function, whereas the associations between the alternative eGFR difference indices and

**Table 1. Baseline Characteristics of Participants.**

| Variable[a] | Baseline eGFRdiff (mL/min/1.73 m²) | | | Overall | P-value |
|---|---|---|---|---|---|
| | **<−15** | **−15–15** | **≥15** | | |
| | **(N = 114925)** | **(N = 320404)** | **(N = 23339)** | **(N = 458668)** | |
| **Mean Age (SD), years** | 58.4 (7.59) | 56.1 (8.12) | 53.1 (8.00) | 56.5 (8.09) | <0.001 |
| **Ethnicity (White, %)** | 108788 (94.7%) | 304591 (95.1%) | 20749 (88.9%) | 434128 (94.6%) | <0.001 |
| **Sex (Female, %)** | 58907 (51.3%) | 176673 (55.1%) | 13662 (58.5%) | 249242 (54.3%) | <0.001 |
| **Education** | | | | | |
| college or above | 28910 (25.2%) | 50116 (15.6%) | 2520 (10.8%) | 81546 (17.8%) | <0.001 |
| high school or equivalent | 56026 (48.8%) | 159409 (49.8%) | 12079 (51.8%) | 227514 (49.6%) | |
| less than high school | 29989 (26.1%) | 110879 (34.6%) | 8740 (37.4%) | 149608 (32.6%) | |
| **Non-smokers (%)** | 64238 (55.9%) | 145937 (45.5%) | 9330 (40.0%) | 219505 (47.9%) | <0.001 |
| **Alcohol consumption (%)** | | | | | |
| Never | 7091 (6.2%) | 11816 (3.7%) | 746 (3.2%) | 19653 (4.3%) | <0.001 |
| Previous | 6200 (5.4%) | 9532 (3.0%) | 528 (2.3%) | 16260 (3.5%) | |
| Current | 97276 (84.6%) | 280907 (87.7%) | 20602 (88.3%) | 398785 (86.9%) | |
| Current, excessive | 4358 (3.8%) | 18149 (5.7%) | 1463 (6.3%) | 23970 (5.2%) | |
| **Physical activity (MET-min/week)** | | | | | |
| High | 36948 (32.1%) | 107119 (33.4%) | 8043 (34.5%) | 152110 (33.2%) | <0.001 |
| Moderate | 55311 (48.1%) | 163290 (51.0%) | 12024 (51.5%) | 230625 (50.3%) | |
| Low | 22666 (19.7%) | 49995 (15.6%) | 3272 (14.0%) | 75933 (16.6%) | |
| **Social isolate (%)** | 24772 (21.6%) | 69766 (21.8%) | 5516 (23.6%) | 100054 (21.8%) | 0.123 |
| **Median TDI (IQR)** | −1.69 (−3.40 to 1.30) | −2.31 (−3.73 to 0.160) | −2.26 (−3.70 to 0.300) | −2.17 (−3.66 to 0.470) | <0.001 |
| **Cho (>5.7mmol/L)** | 55724 (48.5%) | 155709 (48.6%) | 10153 (43.5%) | 221586 (48.3%) | 0.523 |
| **LDL-C (>2.6mmol/L)** | 98793 (86.0%) | 278597 (87.0%) | 19932 (85.4%) | 397322 (86.6%) | <0.001 |
| **Hearing loss (%)** | 13315 (11.6%) | 37420 (11.7%) | 2348 (10.1%) | 53083 (11.6%) | 0.401 |
| **Vision Impairment (%)** | 22260 (19.4%) | 46998 (14.7%) | 2473 (10.6%) | 71731 (15.6%) | <0.001 |
| **NCDs** | | | | | |
| Depression (%) | 27897 (24.3%) | 69818 (21.8%) | 5115 (21.9%) | 102830 (22.4%) | <0.001 |
| Obesity (%) | 43984 (38.3%) | 63838 (19.9%) | 3335 (14.3%) | 111157 (24.2%) | <0.001 |
| Diabetes (%) | 105815 (92.1%) | 306477 (95.7%) | 22694 (97.2%) | 434986 (94.8%) | <0.001 |
| Hypertension (%) | 80417 (70.0%) | 184073 (57.5%) | 11193 (48.0%) | 275683 (60.1%) | <0.001 |
| **Mean eGFRcr_cys (SD), mL/min/1.73 m²** | 84.9 (14.5) | 96.9 (18.3) | 97.8 (17.2) | 94.0 (18.1) | <0.001 |
| **Mean eGFRcr (SD), mL/min/1.73 m²** | 97.2 (10.3) | 94.8 (13.5) | 79.8 (11.7) | 94.6 (13.1) | <0.001 |
| **Mean eGFRcys (SD), mL/min/1.73 m²** | 73.9 (11.6) | 92.5 (14.5) | 102 (11.0) | 88.3 (16.2) | <0.001 |
| **Mean eGFRdiff (SD), mL/min/1.73 m²** | −23.3 (6.92) | −2.33 (7.40) | 22.3 (6.94) | −6.32 (13.3) | <0.001 |
| **Mean eGFRratio (SD)** | 0.759 (0.0727) | 0.976 (0.0835) | 1.29 (0.132) | 0.938 (0.150) | <0.001 |

eGFRcr_cys, estimated glomerular filtration rate calculated using both creatinine and cystatin C; eGFRcr, creatinine-based estimated glomerular filtration rate; eGFRcys, cystatin C–based estimated glomerular filtration rate; eGFRdiff; the difference between eGFRcys and eGFRcr; eGFRratio: the ratio between eGFRcys and eGFRcr; eGFRrd: the relative difference between eGFRcys and eGFRcr; TDI, Townsend deprivation index; MET, metabolic equivalent; LDL-C, serum low-density lipoprotein cholesterol; Cho: serum cholesterol; NCDs: Noncommunicable diseases.

[a]The values for categorical variables are given as numbers (%); values for continuous variables are given as median [Inter Quartile Range, IQR] or mean (standard deviation, SD).

**Table 2. Associations between difference in cystatin C- and creatinine-based estimated glomerular filtration rate and incident dementia.**

| Outcome and exposure | Cases (Rate per 100 person years) | Crude hazard ratio (95%CI) | Adjusted hazard ratio[a] (95%CI) |
|---|---|---|---|
| **All cause dementia** | | | |
| Categorical eGFR$_{diff}$ | | | |
| Negative (~−15 ml/min/1.73 m$^2$) | 3087 (2.04) | 1.0 (ref) | 1.0 (ref) |
| Midrange(~ 15 ml/min/1.73 m$^2$) | 5404 (1.24) | 0.60 (0.58 to 0.63) | 0.88 (0.84 to 0.92) |
| Positive (> 15 ml/min/1.73 m$^2$) | 219 (0.69) | 0.33 (0.29 to 0.38) | 0.74 (0.65 to 0.86) |
| Continuous eGFR$_{diff}$ (per SD ml/min/1.73m$^2$) | | 0.73 (0.72 to 0.75) | 0.92 (0.90 to 0.94) |
| **Alzheimers disease** | | | |
| Categorical eGFR$_{diff}$ | | | |
| Negative (~−15 ml/min/1.73 m$^2$) | 1309 (0.86) | 1.0 (ref) | 1.0 (ref) |
| Midrange(~ 15 ml/min/1.73 m$^2$) | 2498 (0.57) | 0.66 (0.61 to 0.70) | 0.92 (0.85 to 0.98) |
| Positive (> 15 ml/min/1.73 m$^2$) | 103 (0.32) | 0.37 (0.30 to 0.45) | 0.80 (0.65 to 0.98) |
| Continuous eGFR$_{diff}$ (per SD ml/min/1.73m$^2$) | | 0.76 (0.74 to 0.79) | 0.94 (0.91 to 0.98) |
| **Vascular dementia** | | | |
| Categorical eGFR$_{diff}$ | | | |
| Negative (~−15 ml/min/1.73 m$^2$) | 733 (0.48) | 1.0 (ref) | 1.0 (ref) |
| Midrange(~ 15 ml/min/1.73 m$^2$) | 1121 (0.26) | 0.53 (0.48 to 0.58) | 0.85 (0.77 to 0.94) |
| Positive (> 15 ml/min/1.73 m$^2$) | 39 (0.12) | 0.25 (0.18 to 0.34) | 0.67 (0.49 to 0.93) |
| Continuous eGFRdiff (per SD ml/min/1.73m$^2$) | | 0.67 (0.64 to 0.71) | 0.90 (0.86 to 0.95) |

HR, hazard ratio; CI, confidence interval; AD, Alzheimer's disease; VaD, vascular dementia; eGFRcr_cys, estimated glomerular filtration rate calculated using both creatinine and cystatin C; eGFRdiff, the difference between creatinine-based estimated glomerular filtration rate; TDI, Townsend deprivation index; MET, metabolic equivalent; LDL-C, serum low-density lipoprotein cholesterol; Cho: serum cholesterol; NCDs: Noncommunicable

[a] Age-scaled models were adjusted for eGFRcr_cys, Cho, LDL, education, smoking, drinking, physical activities, Townsend deprivation index (TDI), social isolate, hearing, eyesight, diabetes, hypertension, depression, and obesity. diseases.

dementia incidence remained consistent (S5 Fig, S4-S6 Tables in S1 File). Additionally, a negative-control analysis using injury/poisoning codes (S7 Table in S1 File) showed no association with eGFR$_{diff}$, further supporting the specificity of our findings.

**Optimal cut-off determination and risk Stratification of eGFR$_{diff}$**

Using maximally selected rank statistics, the optimal cut-off value of the eGFR$_{diff}$-based score was identified as −8.813. Participants with eGFR$_{diff}$ values ≤ −8.813 were classified as high risk, whereas those with values > −8.813 were classified as low risk. Kaplan–Meier curves demonstrated a significantly higher cumulative incidence of all-cause dementia in the high-risk group compared with the low-risk group (log-rank p < 0.001) (S6 Fig in S1 File).

Time-dependent ROC analyses showed good and increasing discrimination over follow-up for both models (S7 Fig in S1 File). At 5 years, the AUC was 0.801 for the UKBDRS model alone (Model 1) and increased to 0.810 after adding eGFR$_{diff}$ (Model 2). At 10 years, the AUC increased from 0.821 in Model 1 to 0.825 in Model 2. At 15 years, the AUC further increased from 0.848 in Model 1 to 0.850 in Model 2. Overall, the addition of eGFR$_{diff}$ to UKBDRS resulted in a consistent but modest improvement in long-term discriminatory performance.

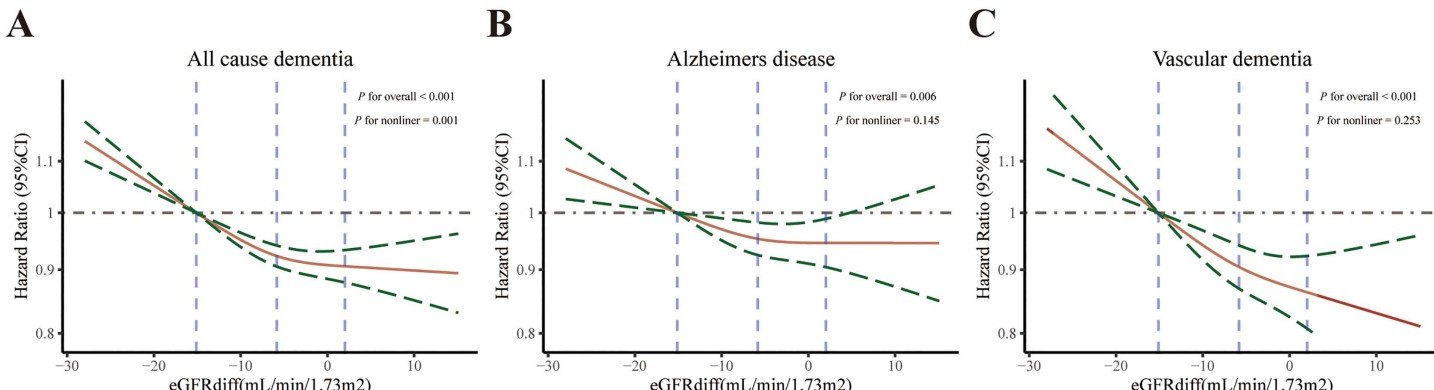

**Fig 1. Dose-response relationship between eGFRdiff and All cause dementia(A), Alzheimers disease(B), or Vascular dementia(C).** Restricted cubic spline was used to explore nonlinear associations, with three knots fixed at the quartiles for all smooth curves. Green line representing 95% Confidence interval. The HR was derived using Cox proportional hazard regression. Model were adjusted for eGFRcr_cys, Cho, LDL, education, smoking, drinking, physical activities, Townsend deprivation index (TDI), social isolate, hearing, eyesight, diabetes, hypertension, depression, and obesity.

### Association of dementia with neuroimaging changes and cognitive functions

A higher z-score of the $eGFR_{diff}$ was significantly associated with greater total brain (β = 0.010, 95% CI, 0.001–0.019) and gray matter (β = 0.024, 95% CI, 0.016–0.033) volumes and lower WMH volumes (β = −0.041, 95% CI, −0.051 to −0.031). However, no significant associations were detected between the $eGFR_{diff}$ and hippocampal volume, whether left (β = −0.00, 95% CI, −0.01 to 0.01) or right hippocampal volume (β −0.00, 95% CI, −0.01 to 0.01). For cognitive function, the z-score of $eGFR_{diff}$ showed no significant correlations with the Trail Making Test A (β = −0.011, 95% CI, −0.020 to 0.001), whereas a slight negative correlation was observed for the Trail Making Test B (β = −0.015, 95% CI, −0.024 to 0.005). In contrast, a slight positive correlation was observed between the $eGFR_{diff}$ z-score and performance in several cognitive domains, including verbal and numerical reasoning (β = 0.019, 95% CI, 0.014–0.024), working memory (β = 0.025, 95% CI, 0.015–0.034), complex processing speed (β = 0.031, 95% CI, 0.022–0.039), verbal declarative memory (β = 0.025, 95% CI, 0.016–0.035), and non-verbal reasoning (β = 0.014, 95% CI, 0.005–0.023) (Table 3).

## Discussion

### Summary of findings

This large, population-based prospective cohort study revealed significant associations between the intraindividual $eGFR_{diff}$ and the risk of dementia, as well as neuroimaging changes and cognitive functions. An L-shaped relationship was observed between the $eGFR_{diff}$ and ACD risk. The observed temporal sequence—baseline $eGFR_{diff}$ preceding MRI markers—supports a directional association. Indeed, negative $eGFR_{diff}$ was associated with reduced total brain and gray matter volumes, increased white matter hyperintensities, and worse cognitive performance in dementia-free participants. These alterations may reflect early manifestations of brain vulnerability [22]. Subgroup analyses revealed consistent associations across various demographic and genetic factors, with notable modifications observed according to APOE genotype and ethnicity. These findings highlight the potential of the $eGFR_{diff}$ as an indicator of dementia.

### Interpretation of the findings

These findings are consistent with previous research on the associations between reduced $eGFR_{diff}$ and adverse clinical outcomes, including frailty, depression, sarcopenia, cardiovascular events, and mortality in cohorts such as the Systolic Blood Pressure Intervention Trial (SPRINT), the Cardiovascular Health Study (CHS), and the UKB [16,18,20,21,39]. An

**A**

**All−cause dementia**

| Variable | Count | Percent | | HR (95%CI) | P value | P for interaction |
|---|---|---|---|---|---|---|
| Overall | 458668 | 100 | | 0.93 (0.90 to 0.95) | <0.001 | |
| APOE_genotype | | | | | | 0.01 |
| non−e4 carriers | 335956 | 74.1 | | 0.91 (0.88 to 0.95) | <0.001 | |
| e4 carriers | 117668 | 25.9 | | 0.93 (0.90 to 0.97) | <0.001 | |
| PRS.Q | | | | | | 0.061 |
| PRS low | 226857 | 50 | | 0.92 (0.88 to 0.95) | <0.001 | |
| PRS high | 226767 | 50 | | 0.94 (0.91 to 0.97) | <0.001 | |
| sex | | | | | | 0.494 |
| female | 249242 | 54.3 | | 0.95 (0.92 to 0.98) | 0.005 | |
| male | 209426 | 45.7 | | 0.91 (0.88 to 0.94) | <0.001 | |
| ethnic | | | | | | 0.034 |
| Others | 24540 | 5.4 | | 1.04 (0.94 to 1.14) | 0.476 | |
| White | 434128 | 94.6 | | 0.92 (0.90 to 0.94) | <0.001 | |

0.7  0.8  0.9  1  1.1 1.2 1.3

**B**

**Alzheimer's disease**

| Variable | Count | Percent | | HR (95%CI) | P value | P for interaction |
|---|---|---|---|---|---|---|
| Overall | 458668 | 100 | | 0.95 (0.92 to 0.99) | 0.006 | |
| APOE_genotype | | | | | | 0.066 |
| non−e4 carriers | 335956 | 74.1 | | 0.94 (0.89 to 0.99) | 0.026 | |
| e4 carriers | 117668 | 25.9 | | 0.96 (0.91 to 1.00) | 0.066 | |
| PRS.Q | | | | | | 0.461 |
| PRS low | 226857 | 50 | | 0.95 (0.89 to 1.00) | 0.069 | |
| PRS high | 226767 | 50 | | 0.96 (0.91 to 1.00) | 0.067 | |
| sex | | | | | | 0.355 |
| female | 249242 | 54.3 | | 0.95 (0.90 to 1.00) | 0.047 | |
| male | 209426 | 45.7 | | 0.96 (0.91 to 1.01) | 0.085 | |
| ethnic | | | | | | 0.219 |
| Others | 24540 | 5.4 | | 1.08 (0.92 to 1.25) | 0.346 | |
| White | 434128 | 94.6 | | 0.94 (0.91 to 0.98) | 0.002 | |

0.7  0.8  0.9  1  1.1 1.2 1.3

**C**

**Vascular dementia**

| Variable | Count | Percent | | HR (95%CI) | P value | P for interaction |
|---|---|---|---|---|---|---|
| Overall | 458668 | 100 | | 0.91 (0.86 to 0.95) | <0.001 | |
| APOE_genotype | | | | | | 0.338 |
| non−e4 carriers | 335956 | 74.1 | | 0.90 (0.84 to 0.96) | 0.002 | |
| e4 carriers | 117668 | 25.9 | | 0.91 (0.84 to 0.98) | 0.014 | |
| PRS.Q | | | | | | 0.725 |
| PRS low | 226857 | 50 | | 0.93 (0.87 to 1.01) | 0.075 | |
| PRS high | 226767 | 50 | | 0.88 (0.83 to 0.95) | <0.001 | |
| sex | | | | | | 0.085 |
| female | 249242 | 54.3 | | 0.96 (0.89 to 1.04) | 0.319 | |
| male | 209426 | 45.7 | | 0.87 (0.82 to 0.93) | <0.001 | |
| ethnic | | | | | | 0.003 |
| Others | 24540 | 5.4 | | 1.29 (1.03 to 1.60) | 0.023 | |
| White | 434128 | 94.6 | | 0.89 (0.84 to 0.93) | <0.001 | |

0.7  0.8  0.9  1  1.1 1.2 1.3

**Fig 2. Subgroup analyses of the associations of eGFRdiff with all-cause dementia(A), AD(B) and VaD(C).** Model were adjusted for eGFRcr_cys Cho, LDL, education, smoking, drinking, physical activities, Townsend deprivation index (TDI), social isolate, hearing, eyesight, diabetes, hypertension, depression, and obesity.

**Table 3. Association of z-score of difference in cystatin C- and creatinine-based estimated glomerular filtration rate with neuroimaging outcomes and cognitive function.**

| Z-score of Outcomes | Number | β (95%CI)[a] | p-value |
|---|---|---|---|
| **Neuroimaging Measures** | | | |
| Total Brain Volume | 38596 | 0.010 (0.001 to 0.019) | 0.037 |
| Grey Matter Volume | 38596 | 0.024 (0.016 to 0.033) | <0.001 |
| White Matter Volume | 38596 | −0.011 (−0.022 to 0.000) | 0.053 |
| Left Hippocampal Volume | 38581 | 0.005 (−0.006 to 0.016) | 0.351 |
| Right Hippocampal Volume | 38581 | 0.005 (−0.005 to 0.016) | 0.324 |
| White Matter Hyperintensity Volume | 37442 | −0.041 (−0.051 to −0.031) | <0.001 |
| **Cognitive Test Measures** | | | |
| Executive Function (Trail Making Test A) | 49910 | −0.011 (−0.020 to −0.001) | 0.029 |
| Executive Function (Trail Making Test B) | 49910 | −0.015 (−0.024 to −0.005) | 0.003 |
| Verbal & Numerical Reasoning (Fluid Intelligence) | 148759 | 0.019 (0.014 to 0.024) | <0.001 |
| Working Memory (Backward Digit Span Task) | 47295 | 0.025 (0.015 to 0.034) | <0.001 |
| Complex Processing Speed (Symbol Digit Substitution) | 49518 | 0.031 (0.022 to 0.039) | <0.001 |
| Verbal Declarative Memory (Paired Associate Learning) | 49910 | 0.025 (0.016 to 0.035) | <0.001 |
| Non Verbal Reasoning (Matrix Pattern Completion) | 49461 | 0.014 (0.005 to 0.023) | 0.003 |

CI, confidence interval; $eGFR_{cr\_cys}$, estimated glomerular filtration rate calculated using both creatinine and cystatin C; $eGFR_{diff}$, the difference between creatinine-based estimated glomerular filtration rate; TDI, Townsend deprivation index; MET, metabolic equivalent; LDL-C, serum low-density lipoprotein cholesterol; Cho: serum cholesterol;

[a] Imaging-related confounds (head size, headmotion, head and table position, and imaging center) were regressed out from the neuroimaging outcomes (whitematter hyperintensity[WMH] volumes were log-transformed). Models were adjusted for eGFRcr_cys, Age, Cho, LDL, education, smoking, drinking, physical activities, Townsend deprivation index (TDI), social isolate, hearing, eyesight, diabetes, hypertension, depression, and obesity.

L-shaped association between $eGFR_{diff}$ and cognitive decline has also been reported in the CHARLS cohort, as well as for diabetic microvascular complications and depressive symptoms [17,19,40]. Our results extend this evidence by demonstrating that negative $eGFR_{diff}$ is associated with higher risks of ACD, AD, and VaD, whereas positive $eGFR_{diff}$ does not confer additional protection beyond the midrange level.

Potential mechanisms may differ by dementia subtype. In AD, $eGFR_{diff}$ may partly reflect muscle-related processes, as prior studies have shown strong associations between $eGFR_{diff}$, muscle mass, and strength [16,41], and sarcopenia itself is a known risk factor for dementia [42]. However, $eGFR_{diff}$ alone is not a specific biomarker for sarcopenia [43], suggesting that additional systemic pathways are likely involved. Another possible mechanism is selective renal filtration impairment, such as "shrunken pore syndrome," which may reduce the clearance of middle-molecular-weight proteins, including amyloid-β isoforms [44,45]. For VaD, microvascular dysfunction may be more relevant, supported by the observed association between $eGFR_{diff}$ and white matter hyperintensities, a marker of cerebral small vessel disease [46–48].

Importantly, $eGFR_{diff}$ is strongly influenced by non-renal determinants and does not solely reflect true differences in glomerular filtration. Creatinine is affected by muscle mass and physical activity, whereas cystatin C is influenced by inflammation, adiposity, thyroid dysfunction, and medication use. Therefore, negative $eGFR_{diff}$ may represent a composite of systemic conditions that are themselves risk factors for dementia. From an epidemiological perspective, $eGFR_{diff}$ may function as an integrated marker of overall health status and biological aging rather than a kidney-specific indicator, which may explain its stronger associations with dementia-related outcomes than eGFR based on either biomarker alone. Accordingly, the observed associations likely reflect shared underlying pathophysiological pathways rather than a direct causal effect of renal dysfunction on dementia risk.

While we propose potential biological pathways linking eGFR$_{diff}$ to dementia subtypes, these mechanisms remain speculative. Our observational design limits our ability to draw definitive conclusions about causality or underlying patho-physiology, and experimental studies are needed to validate these hypotheses.

## Clinical implications and recommendations

Several factors suggest that the eGFR$_{diff}$ might be considered a potential associative indicator that warrants further experimental validation, with implications for risk stratification and public health management. Although the improvement in AUC after adding eGFR$_{diff}$ to UKBDRS was modest, this finding is not unexpected given that UKBDRS already demonstrates strong baseline discrimination. Nevertheless, from an epidemiological and mechanistic perspective, the consistent associations of eGFR$_{diff}$ with dementia incidence, neuroimaging changes, and cognitive function suggest that eGFR$_{diff}$ reflects early systemic vulnerability relevant to neurodegeneration, even if its incremental contribution to prediction accuracy is limited. First, eGFR$_{diff}$ is a highly accessible metric, as both the U.S. National Kidney Foundation and the American Society of Nephrology recommend measuring both creatinine and cystatin C, with the eGFR calculated from these metrics being the most accurate estimate [49]. Second, a significant proportion of individuals exhibit substantial discordance between eGFR$_{cr}$ and eGFR$_{cys}$ values. UKB data indicate that over 40% of participants have differences exceeding 20% [13,50]. A 25-year longitudinal study further demonstrated that individuals whose baseline eGFR$_{cys}$ values were more than 30% lower than their corresponding eGFR$_{cr}$ values tended to maintain this discrepancy over time [51]. These differences may be attributed to the influence of muscle mass and medication use on creatinine-based estimates, whereas cystatin C is affected by non-GFR determinants such as inflammation, steroid therapy, and thyroid dysfunction [52].

Given the observed ethnic heterogeneity in effect estimates, particularly the significant interaction for VaD, eGFR$_{diff}$-based risk assessment tools should not be directly implemented in diverse populations without prior validation. The differential associations may reflect varying prevalence of comorbidities, genetic determinants of cystatin C metabolism, or sociocultural factors affecting kidney function across ethnic groups. Future research should prioritize inclusion of underrepresented minorities to ensure equitable translation of these findings.

Given the established role of combined creatinine and cystatin C assessment in clinical nephrology and the high prevalence of discordant estimates in the general population, incorporating eGFR$_{diff}$ into routine assessment may help identify individuals with elevated systemic vulnerability who may benefit from closer cognitive monitoring and early preventive strategies.

## Limitations

This study has several limitations. First, the observational nature of the study precludes direct causal inferences, and the predominantly white study population limits the generalizability of findings to other racial groups and younger individuals. Second, The study population was predominantly of White ethnicity (94.6%), reflecting the demographic composition of the UK Biobank cohort. This significant limitation restricts the generalizability of our findings to more ethnically diverse populations. Our findings should be interpreted with caution when applied to non-White populations, and validation in diverse cohorts is essential before considering clinical implementation in heterogeneous populations. Third, reliance on hospital records for dementia diagnosis may affect the sensitivity and specificity of case identification. Fourth, because eGFRdiff is substantially affected by non-renal factors, it may capture overall health status rather than isolated kidney function. Although this characteristic supports its use as a global risk marker, it limits mechanistic interpretation and precludes causal inference regarding renal pathways in dementia development. Fifth, the optimal eGFR$_{diff}$ threshold reported herein is data-driven and exploratory; its generalizability to unselected populations and its clinical utility beyond risk stratification remain to be established. Because of the observational nature of this study, these findings support but do not prove a causal role of eGFR$_{diff}$ in dementia pathogenesis. Despite these limitations, this study provides valuable insights into the potential role of the eGFR$_{diff}$ as an early indicator of dementia risk, neuroimaging abnormalities, and cognitive decline..

## Conclusion

This study revealed a strong associations between the eGFR$_{diff}$ and the risk of dementia, adverse neuroimaging outcomes, and cognitive decline in a predominantly White UK population. Given that a significant proportion of the study population exhibited a negative eGFR$_{diff}$, the eGFR$_{diff}$ might be considered a potential associative indicator for dementia, with implications for risk stratification and public health management. Future research should focus on elucidating the underlying mechanism and evaluating its clinical applicability in dementia prevention.

## Supporting information

**S1 File. S1 Fig. Flow diagram of analyses.** Flow diagram of analyses. [a]Sensitivity analysis was conducted in this population using multiple imputation to account for missing data on the exposure and covariates or removing low-eGFR participations. S2 Fig. Correlation matrix of kidney function markers. Hexbin plot of the relation between eGFR$_{cr}$, eGFR$_{cys}$, and eGFR$_{diff}$ at baseline. (A) Correlation between eGFR$_{cr}$ and eGFR$_{cys}$. (B) Correlation between eGFR$_{cr}$ and eGFR$_{diff}$. (C) Correlation between eGFR$_{cys}$ and eGFR$_{diff}$. S3 Fig. Survival curves and proportional hazards assessment. Kaplan-Meyer survival curves using time-scale and scatter plot of the scaled Schoenfeld residuals for eGFR$_{diff}$ and all cause dementia(A), Alzheimer's disease(B) and vascular dementia(C). S4 Fig. Nonlinear dose-response relationships. Dose-response relationship between eGFR$_{ratio}$ (A to C) and All cause dementia, Alzheimer's disease, or Vascular dementia. Restricted cubic spline was used to explore nonlinear associations, with three knots fixed at the quartiles for all smooth curves. Green line representing 95% Confidence interval. The HR was derived using Cox proportional hazard regression. Model were adjusted for eGFR$_{cr\_cys}$, Cho, LDL, education, smoking, drinking, physical activities, Townsend deprivation index (TDI), social isolate, hearing, eyesight, diabetes, hypertension, depression, and obesity. S5 Fig. Sensitivity analyses of eGFR measures. Associations between eGFR$_{diff}$ or eGFR$_{ratio}$(z-score) and incident dementia. (A) Model 1 were adjusted for Cho, LDL, education, smoking, drinking, physical activities, Townsend deprivation index (TDI), social isolate, hearing, eyesight, diabetes, hypertension, depression, and obesity. (B) Model 2 were further adjusted for eGFR$_{cr\_cys}$. S6 Fig. Risk stratification by optimal eGFRdiff cut-off. Kaplan–Meier curves for incident all-cause dementia according to high- and low-risk groups defined by the optimal cut-off value of eGFRdiff (−8.813) derived from maximally selected rank statistics. S7 Fig. Incremental predictive value of eGFRdiff. Time-dependent ROC curves comparing discrimination performance of UKBDRS alone versus UKBDRS combined with eGFRdiff at 5, 10, and 15 years of follow-up. S1 Table. Association between eGFRdiff and dementia after multiple imputation. Hazard ratios (95% CI) for categorical and continuous eGFRdiff with all-cause dementia, Alzheimer's disease, and vascular dementia following multiple imputation of missing covariates. S2 Table. Association between eGFRdiff and dementia excluding low eGFR participants. Sensitivity analysis showing hazard ratios for eGFRdiff and dementia outcomes after excluding participants with eGFR < 60 ml/min/1.73m². S3 Table. Association between eGFRratio and incident dementia. Hazard ratios for the association between eGFRratio (quartiles and continuous) with all-cause dementia, Alzheimer's disease, and vascular dementia. S4 Table. Stratified analysis in participants with diabetes. Association between eGFRdiff (categorical and continuous) and incident dementia outcomes among participants with diabetes at baseline. S5 Table. Stratified analysis in participants with hypertension. Association between eGFRdiff (categorical and continuous) and incident dementia outcomes among participants with hypertension at baseline. S6 Table. Stratified analysis in participants with depression. Association between eGFRdiff (categorical and continuous) and incident dementia outcomes among participants with depression at baseline. S7 Table. Negative control analysis using traumatic injury. Association between eGFRdiff and incident traumatic injury (ICD-10 S00–T35, T66–78) as a negative control outcome to assess potential residual confounding. S1 Method. Genetic risk assessment methodology and Survival analysis and model discrimination methods. S1 Code. R Code for analyses of this study.
(ZIP)

## Author contributions

**Conceptualization:** Zhiyi Mao.

**Data curation:** Ruilang Lin, Xinyue Guo.

**Formal analysis:** Zhiyi Mao.

**Funding acquisition:** Yongfu Yu.

**Methodology:** Xiaorui Cui.

**Supervision:** Xueying Zheng.

**Validation:** Yuwei Peng.

**Writing – original draft:** Zhiyi Mao, Yuwei Peng.

**Writing – review & editing:** Yongfu Yu, Xueying Zheng.

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
