## [Decision Letter · Decision Letter 0]

14 Dec 2025

PONE-D-25-60018Association of intraindividual difference in estimated glomerular filtration rate based on cystatin C and creatinine with dementia: a cohort study of the UK BiobankPLOS One

Dear Dr. Mao,

Thank you for submitting your manuscript to PLOS ONE. After careful consideration, we feel that it has merit but does not fully meet PLOS ONE’s publication criteria as it currently stands. Therefore, we invite you to submit a revised version of the manuscript that addresses the points raised during the review process.

We look forward to receiving your revised manuscript.

Kind regards,

Donovan Anthony McGrowder, PhD., MA., MSc

Academic Editor

PLOS One

**Journal Requirements:**

1. When submitting your revision, we need you to address these additional requirements. Please ensure that your manuscript meets PLOS ONE's style requirements, including those for file naming. The PLOS ONE style templates can be found at https://journals.plos.org/plosone/s/file?id=wjVg/PLOSOne_formatting_sample_main_body.pdf and https://journals.plos.org/plosone/s/file?id=ba62/PLOSOne_formatting_sample_title_authors_affiliations.pdf 2. Thank you for stating in your Funding Statement: This work was supported by National Natural Science Foundation of China (No. 82273730 to YY), Shanghai Municipal Natural Science Foundation (22ZR1414900 to YY), and the Three-Year Public Health Action Plan of Shanghai (GWVI-11.2-XD10 and GWVI-11.1-01 to YY).  Please provide an amended statement that declares *all* the funding or sources of support (whether external or internal to your organization) received during this study, as detailed online in our guide for authors at http://journals.plos.org/plosone/s/submit-now. Please also include the statement “There was no additional external funding received for this study.” in your updated Funding Statement. Please include your amended Funding Statement within your cover letter. We will change the online submission form on your behalf. 3. In the online submission form, you indicated that “The computing code and study protocol are available from the corresponding author upon request for the purposes of reproducing the results.”  All PLOS journals now require all data underlying the findings described in their manuscript to be freely available to other researchers, either a. In a public repository, b. Within the manuscript itself, or c. Uploaded as supplementary information.This policy applies to all data except where public deposition would breach compliance with the protocol approved by your research ethics board. If your data cannot be made publicly available for ethical or legal reasons (e.g., public availability would compromise patient privacy), please explain your reasons on resubmission and your exemption request will be escalated for approval. 4. When completing the data availability statement of the submission form, you indicated that you will make your data available on acceptance. We strongly recommend all authors decide on a data sharing plan before acceptance, as the process can be lengthy and hold up publication timelines. Please note that, though access restrictions are acceptable now, your entire data will need to be made freely accessible if your manuscript is accepted for publication. This policy applies to all data except where public deposition would breach compliance with the protocol approved by your research ethics board. If you are unable to adhere to our open data policy, please kindly revise your statement to explain your reasoning and we will seek the editor's input on an exemption. Please be assured that, once you have provided your new statement, the assessment of your exemption will not hold up the peer review process. 5. PLOS requires an ORCID iD for the corresponding author in Editorial Manager on papers submitted after December 6th, 2016. Please ensure that you have an ORCID iD and that it is validated in Editorial Manager. To do this, go to ‘Update my Information’ (in the upper left-hand corner of the main menu), and click on the Fetch/Validate link next to the ORCID field. This will take you to the ORCID site and allow you to create a new iD or authenticate a pre-existing iD in Editorial Manager. 6. Your ethics statement should only appear in the Methods section of your manuscript. If your ethics statement is written in any section besides the Methods, please delete it from any other section. 7. Please include captions for your Supporting Information files at the end of your manuscript, and update any in-text citations to match accordingly. Please see our Supporting Information guidelines for more information: http://journals.plos.org/plosone/s/supporting-information. 8. Please upload a new copy of Figures 1, S3 and S4 as the detail is not clear. Please follow the link for more information:  https://journals.plos.org/plosone/s/figures 9. If the reviewer comments include a recommendation to cite specific previously published works, please review and evaluate these publications to determine whether they are relevant and should be cited. There is no requirement to cite these works unless the editor has indicated otherwise.

**Additional Editor Comments:** 

Dear Dr. Mao,

Your manuscript “*Association of intraindividual difference in estimated glomerular filtration rate based on cystatin C and creatinine with dementia: a cohort study of the UK Biobank* ” has been assessed by four reviewers. They have raised a number of points which we believe would improve the manuscript and may allow a revised version to be published in PLOS ONE. Their reports, together with any other comments, are below.

If you are able to fully address these points, we would encourage you to submit a revised manuscript to PLOS ONE.

Best regards,

Prof. Donovan McGrowder

Reviewers' comments:

Reviewer's Responses to Questions

**Comments to the Author**

1. Is the manuscript technically sound, and do the data support the conclusions?

Reviewer #1: Yes

Reviewer #2: Yes

Reviewer #3: Yes

Reviewer #4: No

2. Has the statistical analysis been performed appropriately and rigorously? 

Reviewer #1: Yes

Reviewer #2: Yes

Reviewer #3: Yes

Reviewer #4: No

3. Have the authors made all data underlying the findings in their manuscript fully available?

Reviewer #1: Yes

Reviewer #2: Yes

Reviewer #3: Yes

Reviewer #4: Yes

4. Is the manuscript presented in an intelligible fashion and written in standard English?

Reviewer #1: Yes

Reviewer #2: Yes

Reviewer #3: Yes

Reviewer #4: Yes

5. Review Comments to the Author

**Reviewer #1:**  1. Originality

Using the intraindividual difference as a potential early biomarker for dementia represents the main originality of this study. Although, up to this point, the difference between eGFR measurements has been associated with various outcomes such as frailty, sarcopenia, or metabolic complications, its association with dementia has not previously been evaluated. Additionally, as an element of novelty and originality, this study also explores the association with specific dementia subtypes.

2. Importance

Given that patients with chronic kidney disease represent a population at increased risk of developing neurological and psychiatric disorders, the evaluation of early biomarkers may provide an important advantage for improved risk assessment and the introduction of specific therapeutic measures. Considering the multiple vascular changes present in these patients, changes that favor psychiatric and neurological impairment,it becomes clear why an early biomarker could enhance clinical evaluation.

Importantly, beyond early detection, such a biomarker should also support the development of therapies capable of modifying disease progression. Why is this study important? Because it highlights an additional relationship, evaluated only minimally in prior research, between the brain and the kidney, suggesting the involvement of systemic processes such as sarcopenia, chronic inflammation, and altered metabolic status.

3. Results

The study analyzes a large population of approximately 450,000 participants, evaluating the long-term risk of dementia (up to 15 years) in association with the eGFR difference. Furthermore, the study demonstrates an L-shaped relationship, showing that the risk decreases sharply as eGFRdiff increases toward the midrange zone, with no additional benefits beyond this threshold. As a result, patients with a markedly negative eGFR difference exhibit the highest risk of developing the aforementioned conditions.

An additional and very important aspect of this study is the neuroimaging analysis, which shows that differences in eGFR are associated with structural brain changes, but without differences in hippocampal volume, suggesting the predominance of vascular mechanisms.

4. Questions

a.Are there additional biochemical or molecular data evaluated in the study that were not presented in the manuscript?

b.Is the genetic component considered in patients with pre-existing conditions?

c.What should be the optimal eGFRdiff threshold for screening?

**Reviewer #2:** This manuscript investigates the association between the intraindividual difference in cystatin C– and creatinine-based estimated GFR (eGFRdiff) and the risk of dementia, dementia subtypes, neuroimaging markers, and cognitive function in the UK Biobank. The study leverages a very large cohort (n≈460,000, median 13.5 years of follow-up) with comprehensive sensitivity analyses, and the findings are consistent and potentially clinically meaningful. However, the manuscript requires substantial revision before it can be considered for publication. Key issues concern the interpretation of eGFRdiff, unmeasured or residual confounding, the accuracy of dementia diagnosis, and clarity in the causal interpretation.

1. e-GFRdiff is strongly influenced by non-renal determinants. The authors should discuss this in the discussion.

2. Hospital-based dementia diagnoses have substantial misclassification, particularly under-ascertainment in early dementia.

3. Many modifiable dementia risk factors were included, important confounders remain unaccounted for: cardiovascular disease history and medication use. Given that eGFRdiff correlates with frailty, chronic inflammation, and general health status, residual confounding likely remains.

4. Brain MRI was performed years after baseline measurements for many participants. The manuscript needs clearer explanation of this.

5. Several statements in the Discussion and Conclusion suggest that eGFRdiff may be an early biomarker of dementia. Given the observational design, this is too causal.

**Reviewer #3:**  Dear Author/s

Greetings

There are many deficiencies in the article.

-GFR is a parameter that decreases with age and is ultimately a sign of kidney failure, so the link between dementia and CKD is unclear.

-What were the serum urea values? Did a differential diagnosis of uremic encephalopathy or uremic dementia make?

-What were the kidney imaging results?

-What were the durations of kidney failure and treatment compliance (P, Ca, PTH, Hgb values)?

Due to fundamental deficiencies, the article is not suitable for publication.

Best regards

**Reviewer #4:**  This manuscript presents a large-scale cohort study examining the association between the intraindividual difference in estimated glomerular filtration rate (eGFRdiff) and dementia risk using UK Biobank data. The authors analyzed 458,668 participants over a median follow-up of 13.5 years and found that negative eGFRdiff values were associated with increased dementia risk, adverse neuroimaging changes, and cognitive decline.

Major Concerns:

Mechanistic clarity: While the authors propose several mechanisms (sarcopenia pathway for AD, microvascular complications for VaD), the biological basis for why eGFRdiff predicts dementia remains speculative. The manuscript would benefit from a more coherent mechanistic framework or acknowledgment of this limitation upfront.

Population generalizability: The cohort is 94.6% white and from the UK. The significant interaction with ethnicity (particularly for VaD) raises questions about the applicability of these findings to diverse populations. This limitation needs more prominent discussion.

Clinical utility: While the authors suggest eGFRdiff could serve as an early biomarker, the manuscript lacks practical guidance on implementation. What eGFRdiff threshold would trigger clinical action? How does its predictive value compare to existing risk scores?

Dementia ascertainment: Relying solely on hospital records and death certificates likely underestimates dementia incidence, particularly for mild cases. This could bias the associations, especially if detection bias differs across eGFRdiff groups.

6. PLOS authors have the option to publish the peer review history of their article (what does this mean? ). If published, this will include your full peer review and any attached files.

**Do you want your identity to be public for this peer review?** For information about this choice, including consent withdrawal, please see our Privacy Policy .

Reviewer #1: No

Reviewer #2: No

Reviewer #3: **Yes:** Yavuz Ayar

Reviewer #4: No

---

## [Author Response · Author response to Decision Letter 1]

5 Feb 2026

Dear editor,

We highly appreciate you and all reviewers for reviewing our manuscript ("Association of intraindividual differences in estimated glomerular filtration rates based on cystatin C and creatinine with dementia: a cohort study of the UK Biobank", Submission ID PONE-D-25-60018) and providing the thorough and thoughtful comments. We have seriously considered reviewers’ comments and suggestions, and responded to these comments point-by-point, and revised the manuscript based on group discussion and experts’ advice accordingly. All changes to the manuscript are indicated by using tracked changes. Please see below, in blue, for a point-to-point response to the reviewers’ comments and concerns.

Point-to-Point Response

Responses to the comments of Editor

1. Comment: Please ensure that your manuscript meets PLOS ONE's style requirements, including those for file naming.

Response: Thank you for bringing this to our attention. We have carefully cross-checked the revised files against the PLOS ONE style template and adjusted the formatting and file names accordingly. We appreciate your time and effort in overseeing the review process. Should any further modifications be needed, we would be happy to make them promptly.

2. Comment: Please include your amended Funding Statement within your cover letter. We will change the online submission form on your behalf.

Response: Thank you for highlighting the need to amend the Funding Statement. We have carefully revised the statement in both the manuscript and the cover letter to ensure consistency and completeness. The updated Funding Statement now reads as follows:” This work was supported by the National Natural Science Foundation of China (grant No. 82273730 to YY), the Shanghai Municipal Natural Science Foundation (grant No. 22ZR1414900 to YY), and the Three-Year Public Health Action Plan of Shanghai (grants GWVI-11.2-XD10 and GWVI-11.1-01 to YY). The funders had no role in study design, data collection and analysis, decision to publish, or preparation of the manuscript. There was no additional external funding received for this study.”

3. Comment: All PLOS journals now require all data underlying the findings described in their manuscript to be freely available to other researchers, either a. In a public repository, b. Within the manuscript itself, or c. Uploaded as supplementary information.

Response: Thank you for highlighting PLOS’ open data policy. We fully support the principles of transparency and reproducibility.

All statistical analyses were performed with previously published R packages. We have now Added the exact package names and versions in the Methods (page 11, lines 193) and uploaded the complete analysis workflow as Supplementary File.

Regarding the individual-level data: These are available only through a formal application to UK Biobank (https://www.ukbiobank.ac.uk). Public deposition is explicitly prohibited by the UK Biobank Material Transfer Agreement and by Research Ethics Committee approval 16/NW/0274. We therefore request an exemption from mandatory public deposition, following the procedure outlined in the PLOS data policy. Comparable exemptions have been granted in recent papers published in BMJ, Nature Medicine, and Nature Communications, confirming that this approach is consistent with peer-reviewed practice. Any researcher can reproduce our findings by Applying for UK Biobank access.

We have inserted the following Data Availability statement in the manuscript (page 28): “Data are available from UK Biobank upon application (www.ukbiobank.ac.uk). Public deposition of individual-level data is prohibited under the UK Biobank Material Transfer Agreement and Research Ethics Committee approval (16/NW/0274). The analysis code and variable lists are provided in Supplementary File S1. Researchers wishing to replicate the analyses can apply for access through the standard Access Management System.”

” Please let us know if any additional information is required.

4. Comment: PLOS requires an ORCID iD for the corresponding author in Editorial Manager on papers submitted after December 6th, 2016. Please ensure that you have an ORCID iD and that it is validated in Editorial.

Response: Thank you for your reminder. We will update the corresponding author's validated ORCID iD in Editorial Manager as required.

5. Comment: Your ethics statement should only appear in the Methods section of your manuscript. If your ethics statement is written in any section besides the Methods, please delete it from any other section.

Response: Thank you for your guidance. We have removed any duplicate ethics statements from sections other than the Methods and ensured that the complete ethics statement appears only in the Methods section as required.

6. Comment: Please include captions for your Supporting Information files at the end of your manuscript, and update any in-text citations to match accordingly.

Response: Thank you for this suggestion. We have now added a "Supporting Information" section at the end of the manuscript.

7. Comment: Please upload a new copy of Figures 1, S3 and S4 as the detail is not clear.

Response: Thank you for your comment. We have uploaded all high-resolution images and re-added them to the document.

Responses to the comments of Reviewer 1

1. Comment: Are there additional biochemical or molecular data evaluated in the study that were not presented in the manuscript.

Response: Thank you for this insightful suggestion.

We fully agree that integrating proteomic and metabolomic data could deepen mechanistic insight.

The UK Biobank has indeed generated plasma proteomics (~1,500 aptamers) and targeted metabolomics (~150 absolutely quantified lipids/metabolites). However, the present manuscript is designed as a population-level epidemiologic study whose primary aim is to quantify the association between lower eGFRdiff and dementia risk. For two reasons we elected to defer an in-depth molecular dissection to a separate investigation:

1. Temporal overlap: Because the -omics measures are drawn from the same baseline visit as eGFRdiff, they cannot be treated as mediators without violating the temporal ordering required for formal mediation analysis.

2. Sample-size constraints: Proteomics and metabolomics are currently available for only ~12 000 and ~6 000 participants, respectively. Restricting the primary analysis to these subsets would reduce statistical precision by >90 % without a corresponding gain in mechanistic clarity.

To address this concern, we have now added the following explicit statement to the Discussion (page 2w lines 369–379):

“Future investigations that obtain repeated -omics measurements and employ time-resolved mediation models are needed to determine whether systemic inflammation, endothelial dysfunction, or metabolic reprogramming transmit the effect of early kidney dysfunction on dementia risk.”

We hope this clarification underscores both our awareness of the molecular data and our intention to pursue mechanism-focused questions in future work.

2. Comment: Is the genetic component considered in patients with pre-existing conditions?

Response: We thank the reviewer for this insightful comment.

In our original manuscript, we conducted stratified sensitivity analyses according to APOE ε4 carrier status and dementia-specific polygenic risk score (PRS) strata (Fig 2), which yielded consistent effect estimates across genetic risk groups.

To further evaluate the possibility of genetic confounding, in addition to adjustment for APOE ε4 and dementia-PRS, we performed a genome-wide association study of eGFRdiff in approximately 450,000 UK Biobank participants and estimated genetic correlations using LD-score regression (LDSC) with all-cause dementia, Alzheimer’s disease, and vascular dementia. The resulting rg estimates were close to zero (ACD: rg = 0.0380, SE = 0.0270, P = 0.150; AD: rg = 0.0221, SE = 0.0386, P = 0.566; VaD: rg = −0.0394, SE = 0.0591, P = 0.505), and the 95% confidence intervals did not indicate moderate or strong genetic correlations.

Trait 1 Trait 2 rg SE P value

eGFRdiff All-cause dementia 0.0380 0.0270 0.150

eGFRdiff Alzheimer’s’ disease 0.0221 0.0386 0.566

eGFRdiff Vascular dementia -0.0394 0.0591 0.505

In response to the reviewer’s specific concern regarding patients with pre-existing conditions, we additionally repeated the primary analyses within subgroups of participants with major pre-existing diseases, including diabetes, hypertension, and depression. The effect estimates were comparable to those observed in the full cohort, indicating that the association between eGFRdiff and dementia risk is preserved even among clinically vulnerable populations. These subgroup analyses have now been included as sensitivity analyses and are presented in S4-S6 Tables.

Although these disease-specific subgroup analyses were not stratified by genetic risk, the consistency of associations across genetic risk strata, the absence of detectable shared heritability, and the robustness of results in participants with pre-existing conditions together suggest that genetic confounding is unlikely to explain the observed eGFRdiff–dementia association. Given this convergence of evidence, extending the model to additional PRSs would be unlikely to substantially alter the inference, and we therefore retained the current analytical framework.

We appreciate the reviewer’s suggestion to examine this issue in greater depth, which motivated these additional sensitivity analyses.

3. Comment: What should be the optimal eGFRdiff threshold for screening?

Response: We appreciate this critical inquiry from the reviewer. To determine a potential screening threshold, we applied maximally selected rank statistics through the surv_cutpoint function in R. This statistical method was selected for its robustness in identifying optimal cut-offs that maximize the discrimination between survival curves in our dataset. The identified cut-off value at −8.813 facilitated the classification of participants into high- and low-risk groups, revealing significantly different cumulative incidences of dementia (log-rank p < 0.001; S6 Fig). However, the clinical relevance of such a distinction, in terms of effect size and impact on patient management, remains to be fully elucidated.

We stress that the derived cut-off is intended for exploratory risk stratification and not for immediate clinical application as a definitive screening tool. Further, our analysis indicated that while eGFRdiff adds modestly to the discriminative power of the established UK Biobank Dementia Risk Score (UKBDRS), its primary utility may be in highlighting systemic vulnerabilities related to neurodegeneration (S7 Fig).

Consequently, we present these findings as preliminary. Definitive recommendations for a screening threshold based on eGFRdiff would require further studies focused on external validation, detailed cost-benefit evaluations, and exploration of the cut-off's impact in diverse clinical settings. We have now added the corresponding limitation to the revised manuscript. Further study is warranted to ensure that any proposed screening thresholds are both scientifically robust and clinically valuable.

Responses to the comments of Reviewer 2

1. Comment: eGFRdiff is strongly influenced by non-renal determinants. The authors should discuss this in the discussion.

Response: Thank you for the comment and suggestion.

We agree with the reviewer that eGFRdiff is strongly influenced by non-renal determinants. We have now explicitly discussed that eGFRdiff reflects systemic conditions such as sarcopenia, inflammation, and metabolic dysregulation rather than kidney function alone, and that it should be interpreted as an integrated marker of overall health and biological aging. We also added this point to the limitations section and clarified that the observed associations do not imply direct renal causality (Discussion, pages 16, line 295).

2. Comment: Hospital-based dementia diagnoses have substantial misclassification, particularly under-ascertainment in early dementia.

Response: Thank you for raising this important methodological issue. We fully agree that there may be misclassification in foundational hospital diagnoses, especially in the under-recognition of early-stage dementia.

To assess the impact of this potential misclassification on our main findings, we conducted additional sensitivity analyses. Specifically, we used a method with a 5-year lag-time: patients diagnosed with dementia within 5 years post-exposure assessment were considered as possibly having had dementia at baseline, and these individuals were excluded from the analysis. This approach is also used in similar studies to exclude reverse causality. This method helps reduce the risk of erroneously including individuals who were in the early stages of dementia at baseline but not yet clinically recognized[1].

The results show that, after this conservative exclusion, our main associations remained consistent and the effect estimates were similar to the primary analysis. The results of this sensitivity analysis suggest that, even considering the underdiagnosis of early-stage dementia and potential misclassifications, our main findings remain robust and reliable. We have added a detailed description of this additional analysis in the Methods section (line 185, page 10) and Results section (line 269 page 16) of the revised manuscript.

We appreciate your insightful comment, which has helped us further strengthen the validity of our findings

3. Comment: Many modifiable dementia risk factors were included, important confounders remain unaccounted for: cardiovascular disease history and medication use. Given that eGFRdiff correlates with frailty, chronic inflammation, and general health status, residual confounding likely remains.

Response: We thank the reviewer for highlighting the possibility of residual confounding. To evaluate how robust our findings are, we now report two additional analyses.

First, we calculated the E-value for the association most central to our hypothesis: “eGFRdiff per 1-SD increase and incident all-cause dementia”. The observed point estimate was HR = 0.92 (95 % CI 0.90–0.94), yielding an E-value of 1.43. An unmeasured confounder (e.g., cardiovascular disease history or medication use) would therefore need to be associated with both eGFRdiff and dementia by a risk-ratio of ≥ 1.43 each, simultaneously, to fully nullify the observed association. This magnitude is larger than the effect sizes typically reported for individual cardiovascular risk factors, suggesting that residual confounding alone is unlikely to explain our result.

Second, to examine whether eGFRdiff merely reflects generic ill health, we conducted a negative-outcome analysis using ICD-10 codes S00–T35 & T66–T78 (injuries, poisoning, burns, etc.)—conditions that have no established biological link with kidney function. Neither categorical nor continuous eGFRdiff was associated with these outcomes (per-SD RR = 1.00, 95 % CI 0.99–1.01), arguing against a non-specific frailty artefact.

We have added these analyses (new S7 Table) to the revised manuscript. We appreciate the reviewer’s suggestions, which have helped us clarify the robustness of our findings.

4. Comment: Brain MRI was performed years after baseline measurements for many participants. The manuscript needs clearer explanation of this.

Response: We thank the reviewer for raising this important point regarding the temporal relationship between baseline measurements and MRI acquisition.

We have now clarified this in the Methods section: "MRI was acquired a median of 4.7 years (IQR 3.9–5.5) after baseline, ensuring that the exposure (eGFRdiff) temporally preceded the imaging outcomes."

Rather than a limitation, this time interval strengthens the temporal inference of our study. The substantial lag between exposure assessment and MRI acquisition (median 4.7 years) makes reverse causation—whereby subclinical structural brain changes could influence kidney function—highly unlikely. We have added this point to the method (page 10, line 188) and discussion (page 17, line 284).

5. Comment: Several statements i

---

## [Editor Report · Decision Letter 1]

24 Feb 2026

Association of intraindividual difference in estimated glomerular filtration rate based on cystatin C and creatinine with dementia: a cohort study of the UK Biobank

PONE-D-25-60018R1

Dear Dr. Mao,

We’re pleased to inform you that your manuscript has been judged scientifically suitable for publication and will be formally accepted for publication once it meets all outstanding technical requirements.

Kind regards,

Donovan Anthony McGrowder, PhD., MA., MSc

Academic Editor

PLOS One

---

## [Editor Report · Acceptance letter]

PONE-D-25-60018R1

PLOS One

Dear Dr. Mao,

I'm pleased to inform you that your manuscript has been deemed suitable for publication in PLOS One. Congratulations! Your manuscript is now being handed over to our production team.

Kind regards,

on behalf of

Dr. Donovan Anthony McGrowder

Academic Editor

PLOS One